# Highly-potent, synthetic APOBEC3s restrict HIV-1 through deamination-independent mechanisms

**Mollie M. McDonnell** [1,2,3], **Suzanne C. Karvonen** [2,3], **Amit Gaba** [4], **Ben Flath** [4], **Linda Chelico** [4], **Michael Emerman** [2,3] *

**1** Molecular and Cellular Biology Graduate Program, University of Washington, Seattle, Washington, United States of America, **2** Division of Human Biology, Fred Hutchinson Cancer Research Center, Seattle, Washington, United States of America, **3** Division of Basic Sciences, Fred Hutchinson Cancer Research Center, Seattle, Washington, United States of America, **4** Department of Biochemistry, Microbiology, and Immunology, University of Saskatchewan, Saskatoon, Saskatchewan, Canada

* memerman@fredhutch.org

**Data Availability Statement:** The sequencing reads were uploaded to the NCBI SRA with BioProject accession number PRJNA643546 and PRJNA718082 The computational pipeline used to

## Abstract

The *APOBEC3* (*A3*) genes encode cytidine deaminase proteins with potent antiviral and anti-retroelement activity. This locus is characterized by duplication, recombination, and deletion events that gave rise to the seven *A3*s found in primates. These include three single deaminase domain *A3*s (*A3A*, *A3C*, and *A3H*) and four double deaminase domain *A3*s (*A3B*, *A3D*, *A3F*, and *A3G*). The most potent of the A3 proteins against HIV-1 is A3G. However, it is not clear if double deaminase domain A3s have a generalized functional advantage to restrict HIV-1. In order to test whether superior restriction factors could be created by genetically linking single A3 domains into synthetic double domains, we linked A3C and A3H single domains in novel combinations. We found that A3C/A3H double domains acquired enhanced antiviral activity that is at least as potent, if not better than, A3G. Although these synthetic double domain A3s package into budding virions more efficiently than their respective single domains, this does not fully explain their gain of antiviral potency. The antiviral activity is conferred both by cytidine-deaminase dependent and independent mechanisms, with the latter correlating to an increase in RNA binding affinity. T cell lines expressing this A3C-A3H super restriction factor are able to control replicating HIV-1ΔVif infection to similar levels as A3G. Together, these data show that novel combinations of A3 domains are capable of gaining potent antiviral activity to levels similar to the most potent genome-encoded A3s, via a primarily non-catalytic mechanism.

## Author summary

Antiviral genes are encoded by all organisms to help protect them from viral infections, including proteins encoded by primates to protect them from viruses similar to HIV-1. These antiviral proteins are also called "restriction factors". Some restriction factors are broadly acting, while others are very specific. During the course of evolution, some of

analyze the sequencing data and generate are available on GitHub (https://github.com/molliemcdonnell/SuperRestrictionFactor_Hypermutation2).

**Funding:** This work was supported by a National Science Foundation predoctoral fellowship (NSF DGE-1762114) to M.M.M., National Institutes of Health P50AI150476 (principal investigator [PI], Nevan Krogan; subaward to M.E.), National Institutes of Health R01 AI030927 to M.E., and Canadian Institutes of Health Research grant PJT-162407 to L.C., and a Saskatchewan Health Research Foundation postdoctoral fellowship to A.G. The funders had no role in study design, data collection and analysis, decision to publish, or preparation of the manuscript.

**Competing interests:** The authors have declared that no competing interests exist.

these genes have expanded into multiple copies and rearranged in different versions to give them new activities. In this paper, we validated the hypothesis that one particular antiviral gene family, called the *APOBEC3* family, has the capability of making novel combinations of antiviral human genes with as great, or greater, potency against HIV-1 as the most potent natural member of this family. By combining parts of the APOBEC3 proteins into novel combinations, we created potent antiviral versions that act through a mechanism distinct from existing APOBEC3 proteins.

## Introduction

Positive selection in host antiviral genes is a result of the host-virus "arms-race" due to repeated cycles of host resistance and virus adaptation [1]. These cycles of mutation-selection that increase the evolutionary rate of single amino acid substitutions are characteristic of many host genes that counteract HIV and related lentiviruses [1,2]. Additional innovation in host antiviral genes also occurs through gene duplication and recombination creating antiviral gene families that, through neo- or sub-functionalization, is an attractive evolutionary strategy to expand host anti-pathogen response. For example, most mammals, including humans, encode two paralogs of Mx proteins, MxA and MxB. Human MxA has broad and potent activity against a diverse range of RNA and DNA viruses, while the antiviral scope of human MxB is more limited to lentiviruses and herpesviruses [3–6]. Additionally, *TRIM5*, a potent restriction factor against lentiviruses, is present in only a single copy in most primates, whereas rodents have up to six [7].

Antiviral gene family expansion is also seen within the *apolipoprotein B mRNA editing enzyme catalytic-polypeptide like 3*, *APOBEC3* (shortened to *A3* here) locus. A3s are a family of cytidine deaminases that hypermutate retroviruses, such as HIV-1, as well as endogenous retroelements. The *APOBEC3* (*A3*) locus, which is unique to placental mammals, has undergone dramatic expansion in many mammalian lineages, including primates. For example, the human genome encodes seven *A3* paralogs (named *A3A*, *A3B*, *A3C*, *A3D*, *A3F*, *A3G*, and *A3H*). In the majority of placental mammals, the *A3* loci is flanked by *CBX6* and *CBX7* genes, suggesting that the amplification of *A3* genes has mainly occurred via tandem gene duplication within the locus [8–12]. In addition to this gene duplication, most of the A3 proteins are rapidly evolving in primates, suggesting that each has evolved to counteract pathogens [9,13].

The *A3* gene family encodes a characteristic zinc-coordinating catalytic motif (His-X-Glu-$X_{23-28}$-Pro-Cys-$X_{2-4}$-Cys) which can be grouped into 3 classes (A3Z1, A3Z2, and A3Z3) on the basis of their conserved Z domain sequences. Of the seven A3 paralogs in humans, *A3A*, *A3C*, and *A3H* encode single domain proteins (*A3Z1*, *A3Z2*, and *A3Z3*, respectively), whereas the four remaining A3s are double Z domains. A3B and A3G are categorized as A3Z2-A3Z1, while A3D and A3F are A3Z2-A3Z2 [9,10,12]. The human A3 proteins also vary in their ability to restrict HIV-1. Human A3A and A3B do not have antiviral activity against HIV-1, while human A3G is the most potent naturally found A3 against HIV-1 and other retroviruses [14].

The human *A3* locus has also diversified through polymorphisms that encode proteins with different antiviral activities. For example, the common form of A3C encodes a serine at position 188 and weakly inhibits HIV-1, but a natural variant that encodes for an isoleucine at position 188 has greater antiviral activity [15]. Additionally, A3H has over four major haplotypes circulating in the human population with varying ability to restrict HIV-1 [16–18]. Because of the potent antiviral restriction these A3s pose, lentiviruses, including HIV-1, have evolved to encode an antagonist, Vif, that abrogates restriction by inducing A3 degradation.

Strain-dependent mutations in Vif affect its ability to degrade different A3H variants, indicating that viral polymorphisms also affect A3 activity [19,20].

Despite the *A3* gene variation in their potency, domain composition, and susceptibility to antagonism by Vif, there are combinations of human A3 proteins that remain unsampled. For example, not all of the double Z domain combinations have been sampled in primates, and many combinations of *A3* double domains with polymorphisms are unsampled [12]. We predicted that novel double domain combinations may prove to be more effective inhibitors of HIV-1 and we refer to these kinds of evolutionary-based variants of natural antiviral proteins with improved potency and/or escape from antagonism as "super restriction factors" [21–23]. Our previous study showed that duplicating the single A3Z2 domain protein A3C created an A3C-A3C tandem domain protein with increased antiviral activity relative to its single domain counterpart that was also largely resistant to degradation by HIV-1 Vif [22]. Nonetheless, the gain of antiviral potency of A3C-A3C is relatively modest and not as potent as A3G, which is the most potent human A3 protein so far described against HIV-1.

In this study, we created novel human A3 proteins by combining the single domain A3C with the single domain A3H in different orientations and with different natural polymorphisms and show that these A3C/A3H double domains are at least as potent inhibitors of HIV-1 as A3G. A3C/A3H double domains are packaged into virions more efficiently than their single domain counterparts, but do not have an increase in hypermutation activity relative to their single domain counterparts. Rather, they have gained potent antiviral activities independent of cytidine deamination and have gained stronger affinity for binding RNA. Creation of T cell lines that stably express an A3C/A3H double domain show that A3C-A3H restricts spreading infection of HIV-1ΔVif as well as A3G, albeit again, by a different antiviral mechanism. These studies show that by exploring evolutionary space of the *APOBEC3* locus in humans, novel combinations of poorly active antiviral A3 proteins can be created that are as potent as the most active A3 proteins.

## Results

### A3C/A3H chimeras are at least as potent as A3G

Each of the human A3s is comprised of either one or two of these conserved zinc-coordinating domains: A3Z1, A3Z2, or A3Z3 (Fig 1A). A3H is unique because it is the only A3 with a Z3 domain. Furthermore, in mammals this Z3 domain has been duplicated only in Carnivora, where it subsequently acquired premature stop codons [9] and has not recombined to make a Z3-containing double domain *A3* in any primate genome [12]. However, a splice variant makes a Z2-Z3 fusion protein with antiviral activity in felines has been reported [24]. Therefore, in order to explore the evolutionary potential of novel human A3 combinations, we created synthetic tandem domain proteins consisting of one Z2 and one Z3 domain together in a single protein. These synthetic Z2-Z3 and Z3-Z2 proteins consist of A3H and the common variant of $A3C_{S188}$ domains (Fig 1A). A previous study showed that a human A3C-A3H combination had antiviral activity [24], but here we also used two variants of A3H: haplotype I (hap I), the less stable and less antiviral A3H protein, and haplotype II (hap II), the more stable and more antiviral A3H protein [17,18] (Fig 1A), as well as tested the orientation of the A3 domains in both directions, i.e. both A3C-A3H and A3H-A3C. In addition, the linker between the two domains in our proteins is different, as we designed the Z2/Z3 and Z3/Z2 double domains based on alignments to naturally found double A3Z2 domains, A3D and A3F, and incorporated the short linker sequence between both domains (Arg-Asn-Pro) found in A3D and A3F [22] (see S1 Fig for sequence annotation). These designed Z2/Z3 and Z3/Z2 double

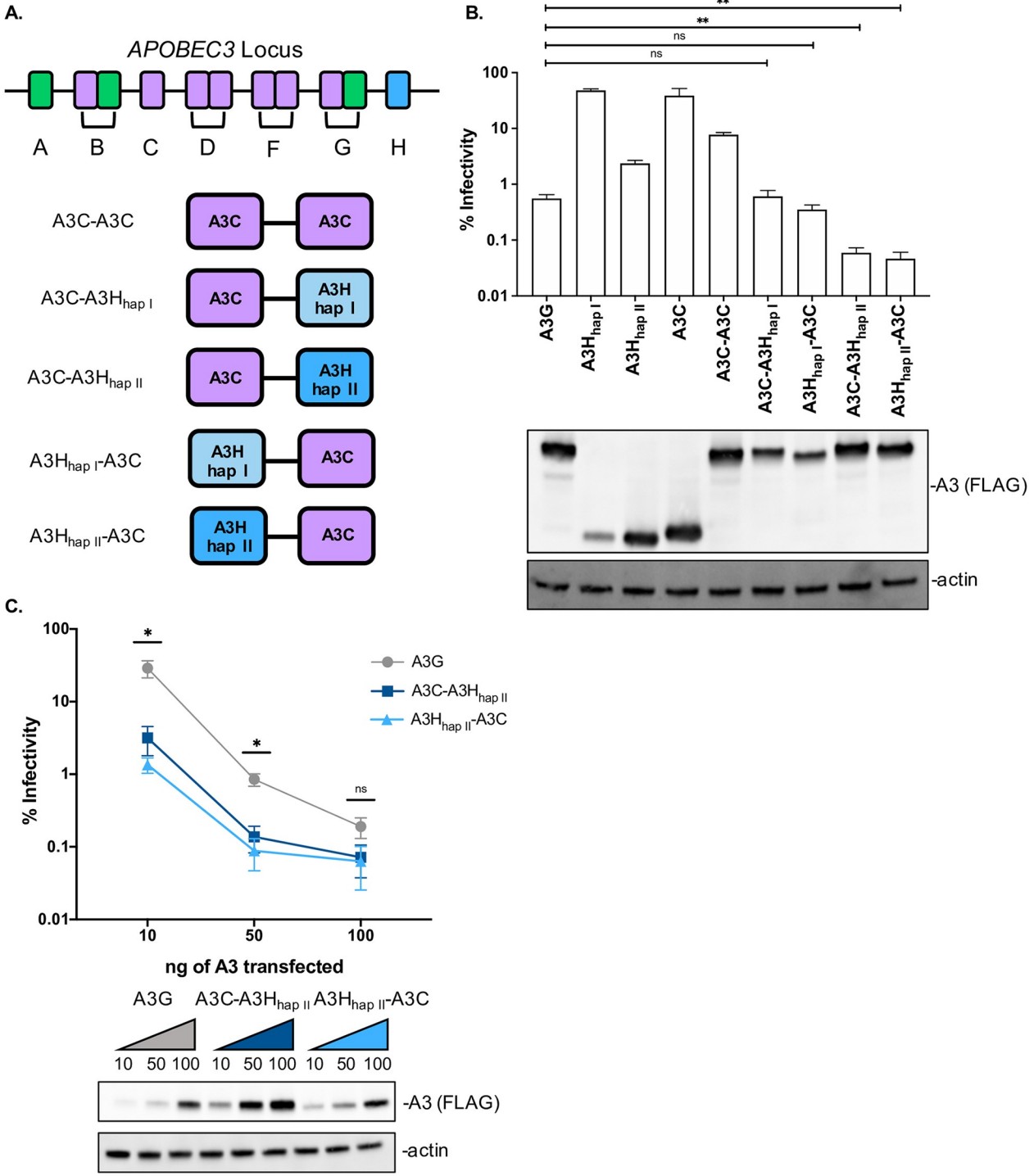

**Fig 1. A3C/A3H double domains are more potent restriction factors than A3G.** (A) Cartoon schematic the *A3* gene locus and of the A3C/A3H double domains synthesized. The Z1 domains are labeled in green, the Z2 domains in purple, and the Z3 domains in blue (light blue for A3H_hap I and dark blue for A3H_hap II). The synthetic double domains, A3C-A3H and A3H-A3C are Z2/Z3 and Z3/Z2, respectively. All double domains used in these experiments have a C-terminal 3XFLAG epitope tag. (B) Top: Single-cycle infectivity assay measuring the percent infectivity of each A3 variant against *HIV-1ΔEnvΔVif* normalized to transfections in the absence of A3. Cells were transfected with 100ng of *A3* and 600ng of *HIV-1ΔEnvΔVif* pseudotyped with 100ng of *VSV-g*. Virus production was quantified by an RT assay (see Methods) and equal amounts of virus was used to infect SUPT1 cells. The bar graph shows an average of 3 biological replicates, each with triplicate infections (+/- SEM). Statistical differences were determined by unpaired *t* tests: ** P≤0.01, ns = not significant. Bottom: Representative western blot of the intracellular levels of A3 in 293Ts. Antibodies to FLAG were used to detect A3s and actin was used as a loading control. (C) Top: The % infectivity of *HIV-1ΔEnvΔVif* pseudotyped

with *VSV-g* and increasing doses of transfected *A3G* (grey), *A3C-A3H$_{hap\ II}$* (dark blue), or *A3H$_{hap\ II}$-A3C* (light blue) are plotted, normalized to a control with no A3. The amount of each *A3* plasmid transfected in ng is shown on the X-axis. Data points are an average of 3 biological replicates, with each biological replicate consisting of 3 triplicate infections (+/- SEM). Statistical differences were determined by unpaired *t* tests between A3G and A3C-A3H$_{hap\ II}$ and A3G and A3H$_{hap\ II}$-A3C: $^{*}$ P $\leq$ 0.05, ns = not significant. Bottom: Western blot showing the intracellular expression levels of A3G, A3C-A3H$_{hap\ II}$, and A3H$_{hap\ II}$-A3C probed with anti-FLAG antibody showing intracellular expression levels for A3s and actin as a loading control. The ng of *A3* transfected are denoted on top of the western blot.

domain A3s are analogous to natural Z2-Z1 and Z2-Z2 combinations that have not yet been naturally sampled in primate lineages.

In order to test the antiviral activity of these proteins, we performed single-cycle infectivity assays by transfecting 293T cells with an expression vector encoding these synthetic genes along with an HIV-1 provirus lacking the A3 antagonist Vif. A3G was used as a positive control, as it is the most potent human A3. A3G can restrict HIV-1ΔEnvΔVif infection by over two orders of magnitude (Fig 1B top). As previously described [17,18], A3H$_{hap\ I}$ weakly inhibits HIV-1ΔEnvΔVif, and A3H$_{hap\ II}$ more potently inhibits HIV-1ΔEnvΔVif, though not as strongly as A3G despite similar expression levels (Fig 1B). A3C also weakly inhibits HIV-1ΔEnvΔVif, but as we previously reported, the antiviral activity of A3C can be increased several fold by creating a synthetic tandem domain, A3C-A3C [22]. Nonetheless, A3C-A3C is still less antiviral than A3G at similar expression levels. In contrast, we found that A3C-A3H$_{hap\ I}$ and A3H$_{hap\ I}$-A3C synthetic tandem domain proteins were as potent A3G (Fig 1B top). Thus, remarkably, two A3 single domain proteins that on their own have little antiviral activity, can produce a synthetic double domain protein with the ability to restrict HIV-1ΔEnvΔVif by two orders of magnitude.

Moreover, when combining A3C with the more active A3H haplotype, A3H$_{hap\ II}$, to create A3C-A3H$_{hap\ II}$ and A3H$_{hap\ II}$-A3C, we could make antiviral proteins that are 9-fold and 11-fold, respectively, more active against HIV-1ΔEnvΔVif than A3G (Fig 1B top). This increase in antiviral activity could not be explained by increased expression levels since A3C and A3H$_{hap\ II}$ single domain proteins are expressed to similar levels as A3C-A3H$_{hap\ II}$ and A3H$_{hap\ II}$-A3C (Fig 1B bottom). Additionally, A3C-A3H$_{hap\ II}$ and A3H$_{hap\ II}$-A3C are expressed to the same level as A3G (Fig 1B bottom).

To more thoroughly examine whether activity is correlated with expression level, we transfected different amounts of plasmids encoding these synthetic tandem domain proteins along with A3G. A3G could restrict HIV-1ΔEnvΔVif approximately 3-fold even at the lowest level of DNA transfected (10ng). However, both A3C-A3H$_{hap\ II}$ and A3H$_{hapII}$-A3C were able to restrict HIV-1ΔEnvΔVif more potently than A3G at every dose tested (Fig 1C). Even at the lowest dose of 10ng with low protein level expression, both A3C-A3H$_{hap\ II}$ and A3H$_{hapII}$-A3C were able to inhibit HIV-1ΔEnvΔVif approximately 30- and 70- fold, respectively. In summary, by creating novel double domains from poorly-restrictive single domain A3s, we can create a super restriction factor that is at least as potent than A3G even at the lowest end of protein expression.

## A3C/A3H chimeras are packaged better than their single domain counterparts

Previous studies have found a direct correlation of increase in packaging to potency of A3s [14,25]. Therefore, we evaluated the packaging of A3C/A3H$_{hap\ II}$ double domains to get packaged into virions. We focused the experiments on A3C-A3H$_{hap\ II}$ and A3H$_{hap\ II}$-A3C (hereafter referred to as A3C-A3H and A3H-A3C) double domains as they were the most potent combination in our assays (Fig 1). The intracellular expression levels of the naturally found A3s, A3G, A3H$_{hap\ II}$, and A3C, are all similar to A3C-A3H and A3H-A3C (Fig 2A top). Both A3C

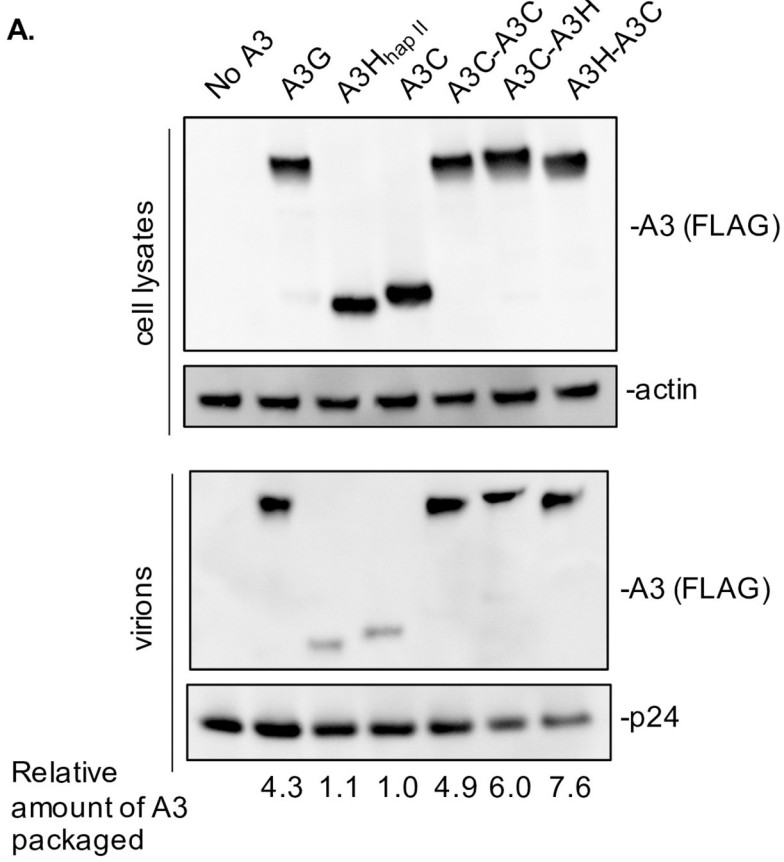

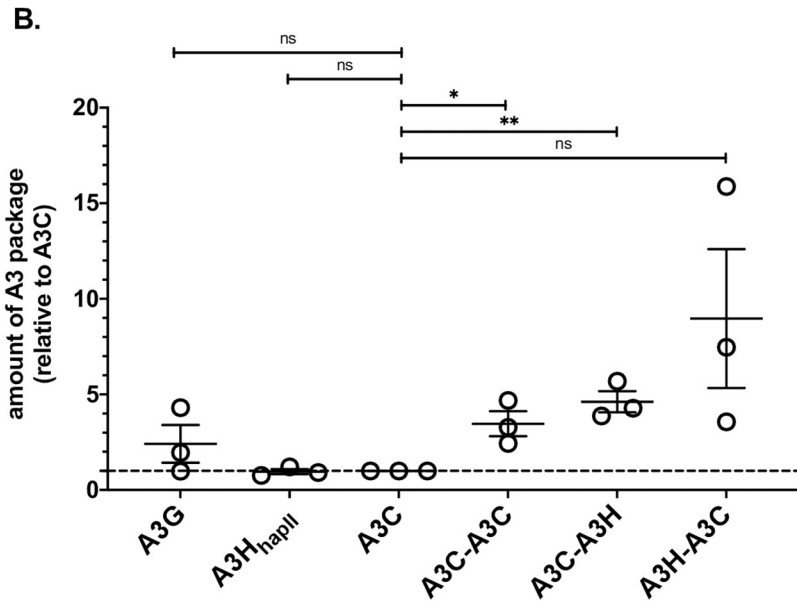

**Fig 2. A3C/A3H double domains are packaged more than their single domain counterparts.** (A) Intracellular expression and packaging of A3 into virions. *HIV-1ΔEnvΔVif* provirus was co-transfected into 293T cells with 100ng of each *A3*. Top: western blot of cellular lysates probed with anti-FLAG antibody showing intracellular expression levels for A3s and actin as a loading control. Bottom: Western blot of proteins in the pelleted virions and probed with anti-FLAG antibody for A3 levels and anti-p24$_{gag}$ for normalization. An empty vector condition was used as a negative

control and labeled no A3. A3C-A3H$_{hap\ II}$ is shortened to A3C-A3H and A3H$_{hap\ II}$-A3C is shortened A3H-A3C. Western blot shown is representative of 3 biological replicates. (B) Quantification of the amount of A3 packaged relative to A3C from three biological replicate transfections. A3 packaged was calculated by dividing the abundance of A3 in the virions normalized to p24$^{gag}$ by the level of A3 expression in the cell normalized to actin. The amount of A3 packaged is reported relative to A3C, whose level was set to a value of 1 and denoted with the dotted line. Error bars represent the SEM. Statistical differences were determined by unpaired $t$ tests between A3C and A3G, A3C and A3C-A3H$_{hap\ II}$, A3C and A3H, and A3C and A3H$_{hap\ II}$-A3C: * P $\leq$ 0.05, ** P $\leq$ 0.01, and ns = not significant.

and A3H$_{hap\ II}$ single domain proteins are poorly incorporated into virions (Fig 2A bottom). As we previously reported [22], the double domain A3C-A3C is packaged 3.3-fold more than A3C and A3G 2.0-fold more than A3C (Fig 2 bottom and quantified from three independent replicates in 2B). Here, we found that A3C/A3H double domains also have an increase in packaging relative to their single domain counterparts; A3C-A3H is packaged 4.6-fold more than A3C and A3H-A3C is packaged 9.0-fold more than A3C (Fig 2 bottom and quantified from three independent replicates in 2B). However, the increase in packaging of A3C/A3H alone is unlikely to explain all of the 650-fold increase in antiviral activity between A3C and A3C/A3H double domains since A3C-A3C is also packaged at similar levels but is not nearly as potent an antiviral protein. Furthermore, the packaging of A3H-A3C is more variable despite still potently restricting HIV-1ΔEnvΔVif in infectivity assays, suggesting that the increase in packaging alone does not completely explain the increase in antiviral activity (Figs 2B and 1B).

## A3C/A3H chimeras have gained a deaminase-independent mechanism to inhibit HIV that correlates with inhibition of reverse transcription (RT) products and increased affinity for RNA

Naturally found A3 proteins primarily use deaminase-dependent mechanisms to inhibit HIV-1 by converting cytidines to uracils on ssDNA in the minus strand of DNA during reverse transcription, leading to G-to-A mutations in the positive strand of proviral DNA [26]. A3G has been documented to induce hypermutation of up to 10% of guanosine residues in the HIV-1 genome [27]. Mutating the active sites in A3G mostly, but not completely, abrogates the antiviral activity, demonstrating the primary uses of deaminase-dependent methods of hypermutation to inhibit HIV-1 replication, although a deaminase-independent mechanism of viral inhibition is still detected [28,29]. Previously, we found that A3C-A3C double domain proteins did not increase the amount of G-to-A mutations in HIV-1 in a single-cycle infectivity assay, but rather increased their antiviral activity through inhibition of reverse transcription [22].

To test whether or not the large increase in antiviral activity of A3C/A3H double domains can be explained by an increase in hypermutation, we analyzed HIV-1 hypermutation induced by each A3C/A3H double domain using a recently described method to deep sequence all G-to-A mutations induced by a given A3 over a region of HIV-1 *pol* [22]. A "plasmid control" was used to identify PCR- and Illumina-induced errors while the "No A3" condition controlled for mutations that arise during reverse transcription (Fig 3A). Consistent with previous results [22], we found that A3G induces more than 1 mutation in over 96% of the reads and more than 10 mutations in over 43% of the reads. In contrast, A3C induced far fewer reads with G-to-A mutations, with only 12% of the reads having 2 or more mutations. As previously reported [22], A3C-A3C induces similar frequencies of G-to-A mutations as A3C. A3H$_{hap\ II}$ induced at least one G-to-A mutation in more than half of all reads and 10 or more mutations in approximately 10% of the reads, demonstrating significant hypermutation, but less than A3G. In contrast, despite the 500-fold increase in antiviral activity of A3C-A3H and A3H-A3C compared to A3H$_{hap\ II}$, we found no increase in hypermutation of A3C-A3H and A3H-A3C

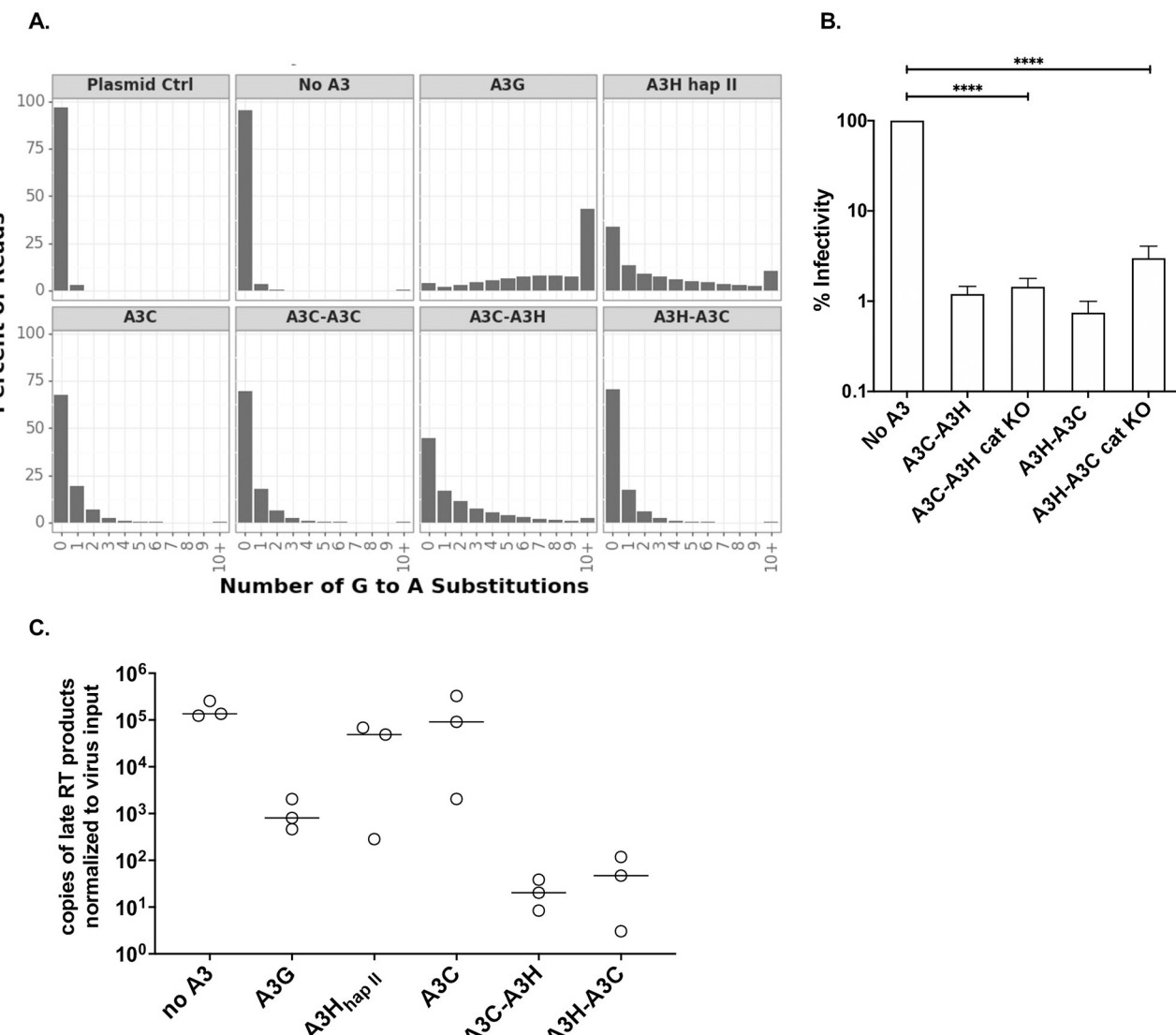

**Fig 3. A3C/A3H double domains use deaminase independent mechanisms to restrict HIV-1.** (A) Paired-end sequencing reads were analyzed for G-to-A mutations. Data is shown as frequency distribution bar graphs of the percent of reads by the number of G-to-A substitutions in each read for each A3 tested. Plasmid control (referred to as plasmid ctrl) was used as a sequencing control and a no A3 sample was used to distinguish background mutations, including reverse transcriptase-induced mutations. A3C-A3H$_{hap\ II}$ is shortened to A3C-A3H and A3H$_{hap\ II}$-A3C is shortened A3H-A3C. (B) Single-cycle infectivity assay measuring the percent infectivity of each A3 variant against *HIV-1ΔEnvΔVif*. Catalytic knockouts of the essential glutamic acid in both N- and C- terminal domains of A3C-A3H$_{hap\ II}$ and A3H$_{hap\ II}$-A3C (shortened to cat KO) were created and compared to their catalytically active counterpart. Cells are transfected with 100ng of *A3* and 600ng of *HIV-1ΔEnvΔVif* pseudotyped with 100ng of *VSV-g*. Virus production was normalized and equal amounts of virus was used to infect SUPT1 cells. Results from each experiment were normalized to a no A3 control. Bar graph shows an average of 3 biological replicates, each with triplicate infections (+/- SEM). Statistical differences were determined by unpaired *t* tests: ns = not significant. (C) To evaluate the relative copies of late reverse transcription products, SUPT1 cells were infected with *HIV-1ΔEnvΔVif* and either *no A3* or 100ng of *A3* to test for inhibition of HIV-1 reverse transcriptase products. 18 hours later, viral cDNA was harvested and the levels of HIV-1 proviral DNA was assayed by qPCR. Each circle represents a normalized value for the respective biological replicate, with qPCR technical duplicates. A3C-A3H$_{hap\ II}$ is shortened to A3C-A3H and A3H$_{hap\ II}$-A3C is shortened A3H-A3C. Each sample has been adjusted for equal viral infection and a nevirapine control. Bars represent the mean across 3 biological replicates.

relative to A3H$_{hap\ II}$ (Fig 3A compare distribution of mutations in right box on the top row with the distribution of mutations in the last right most boxes on the bottom row). A3H-A3C induces at least one mutation in approximately 30% of reads, similar to A3C-A3C or A3C alone. A3C-A3H induces at least one G-to-A mutation in 55% of the reads and 2 or more

mutations in 38% of all reads, similar to the level of A3H$_{hapII}$-mediated hypermutation. The low levels of hypermutation for these potent antiviral double domain proteins suggests a hypermutation-independent mechanism for super restriction.

In order to complement the A3-mediated hypermutation analysis, we also made catalytically inactive A3C/A3H proteins by mutating the glutamic acid essential for the deamination reaction in both domains of the double domain proteins. We found that restriction by the catalytically inactive version of A3C-A3H, A3C-A3H E60A E240A, called A3C-A3H cat KO, was indistinguishable from the unmutated A3C-A3H (0.12% infectivity versus 0.08% infectivity (Fig 3B). This suggests that A3C-A3H primarily uses cytidine deaminase-independent mechanism of restriction, supporting the conclusions from the hypermutation data. Interestingly, in A3H-A3C when both active sites have been mutated to an alanine, A3H-A3C E57A E247A (here called A3H-A3C cat KO), can only inhibit HIV-1ΔEnvΔVif to 0.39% (compared to 0.15% infectivity with wild-type A3H-A3C), suggesting that A3H-A3C also uses both a deaminase-dependent and a deaminase-independent mechanism to restrict HIV-1 (Fig 3B). Thus, these data support a model that novel combinations of A3 domains have created super restriction factors that potently inhibit HIV-1ΔEnvΔVif predominantly through a deaminase-independent mechanism.

We also quantified the amount of late RT products in the presence of synthetic A3s. Unintegrated DNA was harvested 18 hours after infection and quantified by qPCR, normalized to the amount of virus used for each infection. As previously reported [29–31], virus produced in the presence of A3G showed a significant decrease in relative late RT products compared to the no A3 control (Fig 3C). On the other hand, virus produced in the presence of A3H$_{hap II}$ or A3C had similar levels of late RT products as the no A3 control. Strikingly, virus made in the presence of A3C-A3H or A3H-A3C accumulated even fewer late RT products than in the presence of A3G (Fig 3C), mirroring the difference in antiviral activity (Fig 1B). These results suggest that inhibition of the formation of reverse transcriptase products is likely the major mechanism by which the A3C/A3H double domain proteins act and accounts for their greater antiviral activity relative to A3G. These findings support the hypothesis that A3C/A3H super restriction factors function in a novel deaminase-independent mechanism compared to their single domain counterparts.

One possible mechanism for reduced reverse transcriptase products in the presence of A3C-A3H would involve steric hinderance of reverse transcriptase due to competition between the A3 and reverse transcriptase for the template RNA. An increase in affinity for binding to RNA by A3C-A3H is an attractive possibility as A3H has previously been shown to bind RNA [32–35]. To determine the ability of the different A3s to interact with HIV-1 RNA, we conducted steady-state rotational anisotropy with fluorescein labeled HIV 5'UTR RNA and increasing amount of A3. The anisotropy can rise or decrease upon interaction of the binding partners, with a rise indicating a simple interaction and a decrease indicating an interaction and structural change of the 5'UTR [36,37]. The resulting saturation curves were analyzed to determine the dissociation constant ($K_d$), where a lower $K_d$ value indicates less dissociation and tighter binding. We chose to compare A3C-A3H to A3H$_{hap II}$, as previous results found that A3C does not package into virions by binding RNA and does not form RNA mediated multimers in cells, in contrast to A3H [25,34,38]. We found that all the A3s examined decreased the anisotropy of the 5'UTR, suggesting that they were able to change the RNA structure (Fig 4). A3C-A3H bound RNA with a $K_d$ of 0.03 nM, 17-fold stronger than A3H$_{hap II}$ ($K_d = 0.52$ nM), consistent with its increase ability to inhibit RT products (Fig 4). A3G exhibited a much lower affinity for the 5'UTR, with a $K_d$ of 6.36 nM. This is consistent with A3G inhibiting RT at least in part by binding to reverse transcriptase directly [29]. These data support the model that increased affinity for RNA is responsible for an inhibition

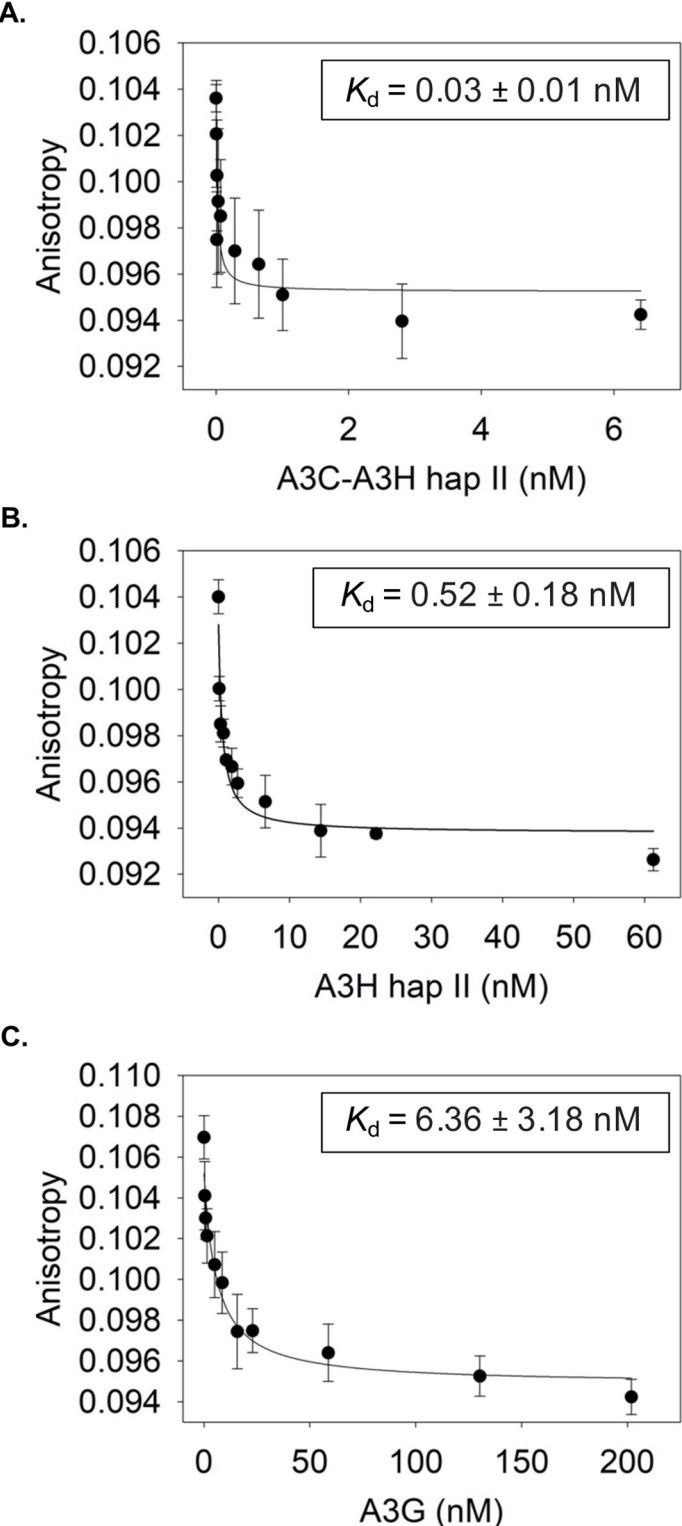

**Fig 4. A3C-A3H has increased binding affinity for the HIV-1 5'UTR.** The apparent $K_d$ of A3 enzymes from the fluorescein labeled 497 nt RNA was analyzed by steady-state rotational anisotropy for (A) A3C-A3H$_{hap\ II}$ (0.03 ± 0.01 nM); (B) A3H$_{hap\ II}$ (0.52 ± 0.18 nM) and (C) A3G (6.36 ± 3.18 nM). The x-axis on each graph is different due to the different amount of protein added in order to fully saturate the RNA. Error bars represent the standard deviation from three independent experiments.

of RT products in the presence of A3C-A3H as a deaminase-independent mechanism of HIV antiviral activity.

## Stable expression of A3C-A3H in T cells inhibits HIV-1 replication in a Vif-dependent manner

All the previous experiments were done as single-cycle infectivity assays after co-transfection of A3 proteins with a provirus to determine antiviral activity. To more closely mimic natural infection, we next tested whether these super restriction factors could also inhibit spreading infections of HIV when stably expressed in a T cell line. We integrated the *A3C-A3H* gene into Jurkat T cells using the Sleeping Beauty transposon, which should integrate a single copy of *A3C-A3H* into the T cell genome [39]. As controls, we also created similar cell lines that express A3G or an empty vector. A3G and A3C-A3H were expressed at similar levels in these lines, as assessed by western blot (Fig 5A).

Jurkat cells expressing empty vector (no A3), A3G, or A3C-A3H were then infected in triplicate at low MOI (at MOI = 0.01 and 0.05) with a replication-competent HIV-1 with a deletion that spans the Vif open reading frame (HIV-1ΔVif). Virus production was monitored over time by collecting supernatant and measuring RT activity using the SG-PERT assay to measure RT in virions in the supernatants (Fig 5B for MOI = 0.01 and S2 Fig for MOI = 0.05) [40]. In Jurkat cells expressing the empty vector, HIV-1ΔVif grew exponentially until peaking at day 10 at both MOIs of infection (Fig 5B and S2 Fig, panel A, "No A3", gray line). There was no initial restriction of HIV-1ΔVif in A3G-expressing cells, as expected given the requirement of packaging before HIV-1 restriction. However, at later time points, HIV-1ΔVif growth was inhibited by the A3G-expressing cells, as the RT levels did not further increase and remained at levels much lower than in Jurkat cells without A3 proteins. Remarkably similar to cells expressing A3G, the cells expressing A3C-A3H also efficiently controlled infection of HIV-1ΔVif after day 5 of infection.

We used the area under the curve (AUC) as a metric to statistically compare virus spreading between cell lines (Fig 5B). We determined the AUC for each of the three biological replicate infections and report the mean and standard error (Fig 5B). We found that the AUC for A3G and A3C-A3H infections were approximately 3-4-fold less than the no A3 Jurkat cells, indicating significant HIV-1 restriction (p = 0.031 and 0.046, respectively, one-way ANOVA and the post hoc Tukey's multiple comparisons test). In contrast, there was no statistical difference between the AUC of the A3G- and A3C-A3H-expressing cells. Thus, these results show that A3C-A3H stably expressed in T cells is as effective as A3G in controlling HIV-1 infection in the absence of Vif.

To determine whether the HIV-1ΔVif inhibition in T cells was due to hypermutation or an alternative mechanism, as determined in the single-cycle infectivity assays (Fig 3), we harvested genomic DNA from infected cells at day 14 (Fig 5B) and deep sequenced a region of *pol* to evaluate if integrated proviruses had signatures of A3 mediated hypermutation (Fig 5C). As in Fig 3, we used a plasmid control to determine background mutations from PCR and Illumina sequencing errors (Fig 5C). As these samples were collected at 14 days post-infection, the no A3 genomic DNA sample contains additional G-to-A mutations compared to the single cycle infectivity assay results in Fig 3A because of the errors made from reverse transcription after multiple rounds of infection [41] (and possibly also due to low level basal A3 proteins in Jurkat cells). In the genomic DNA of the cells expressing A3G, we find that there is an increase in the frequency of G-to-A mutations, with approximately 20% of the reads having 10 or more G-to-A mutations. Consistent with our single-cycle infection data, in the genomic DNA of cells expressing A3C-A3H, we found that there were very few additional G-to-A mutations

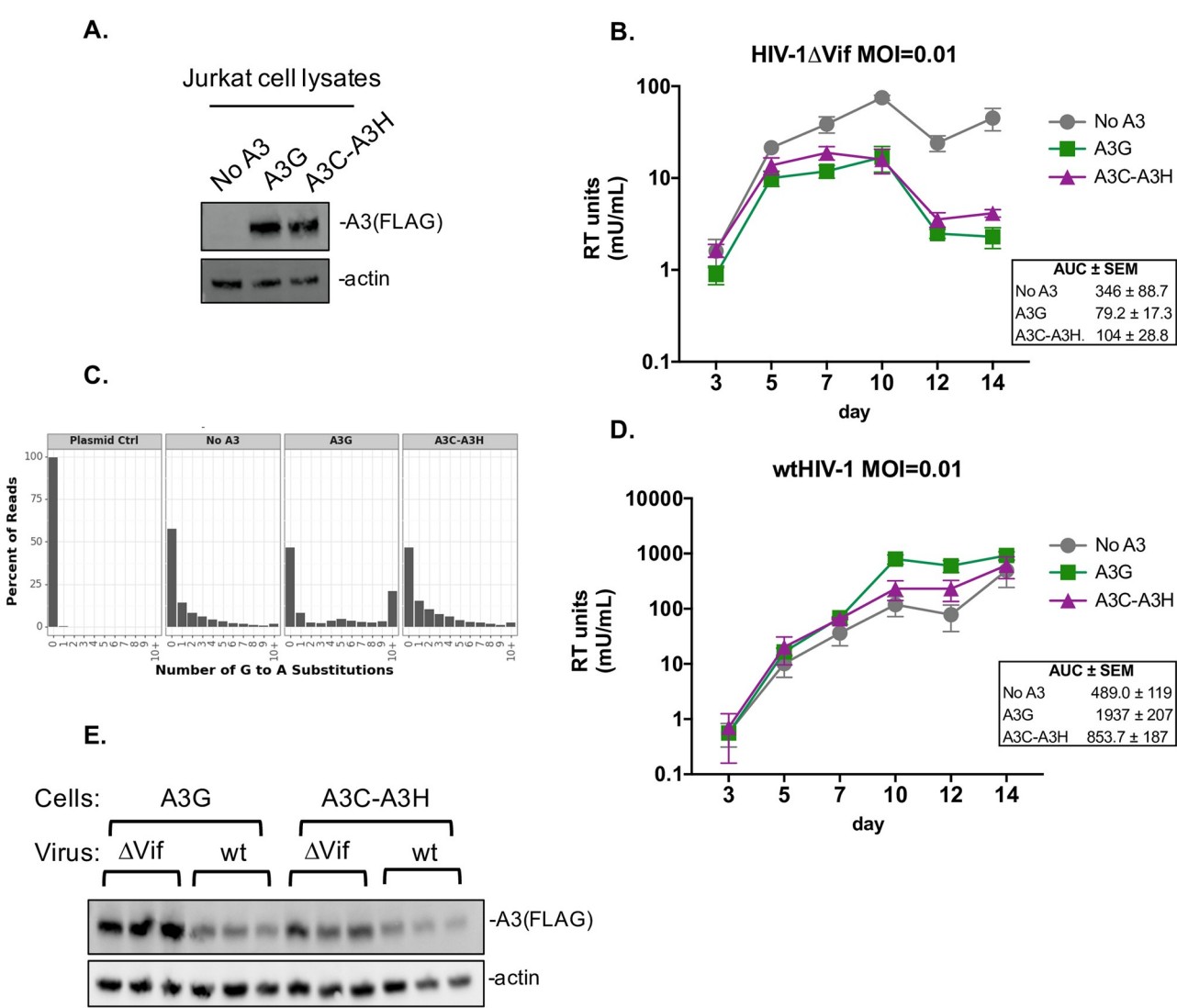

**Fig 5. A3C-A3H suppresses HIV-1ΔVif spreading infection to day 14.** (A) Western blot of the Jurkat cells constitutively expressing no A3, A3G, or A3C-A3H$_{hap\ II}$ (shortened to A3C-A3H) probed with anti-FLAG for the A3 levels and actin as a loading control. (B) Spreading infection kinetics of a replication-competent HIV-1 with a deletion that spans the Vif open reading frame (called HIV-1ΔVif). The Jurkat cells expressing no A3 (circles, grey line), A3G (squares, green line), or A3C-A3H$_{hap\ II}$ (shortened to A3C-A3H, triangles, purple line) were infected at a low MOI (MOI = 0.01) in triplicates. Virus production was monitored over time by collecting supernatant and measuring RT activity (mU/mL) using a SG-PERT assay. Error bars represent the standard error across the 3 biological replicates. To compare spreading infection kinetics, area under the curve (AUC) was calculated for each biological replicate. The mean AUC and standard error of the mean are represented to the right. (C) A3 mediated hypermutation analysis of gDNA from cells harvested on day 14. Paired-end sequencing reads were analyzed for G-to-A mutations. Data is shown as frequency distribution bar graphs of the percent of reads by the number of G-to-A substitutions in each read for each A3 tested. Plasmid control was used as a sequencing control and a no A3 sample was used to distinguish mutations that occurred throughout the 14-day time course. Frequencies are calculated as the average frequency of each biological infection replicates and read counts are shown as the sum of the reads for each replicate. (D) Spreading infection kinetics of a replication-competent wtHIV-1 (LAI isolate). Jurkat cells expressing no A3 (circles, grey line), A3G (squares, green line), and A3C-A3H$_{hap\ II}$ (shortened to A3C-A3H, triangles, purple line) were infected in triplicate at a low MOI (MOI = 0.01). Virus production was monitored over time by collecting supernatant and measuring RT activity (mU/mL) using a SG-PERT assay. Error bars represent the standard error across the 3 biological replicates. To compare spreading infection kinetics, area under the curve (AUC) was calculated for each biological replicate. The mean AUC and standard error of the mean are represented to the right. (E) Western blot of cell lysates collected on day 14 from HIV-1ΔVif infection (B), shortened to ΔVif, and from wtHIV-1 infection (D), shortened to wt. Cells expressing A3G or A3C-A3H$_{hap\ II}$ were evaluated for their intracellular expression levels of A3 in each of the triplicate infections. The anti-FLAG antibody was used to probe for the FLAG tagged A3s and actin was used as a loading control.

when compared to the no A3 cells. In fact, the no A3 and A3C-A3H frequency graphs look nearly identical. These data show that A3C-A3H stably expressed in T cells can inhibit a Vif-deficient HIV-1 as well as A3G, but that the mechanisms of increased inhibition are largely independent of hypermutation.

We also tested if Vif could overcome the antiviral activity A3C-A3H in this system by infecting each of the three cells lines with wtHIV-1 infection (i.e. HIV-1 that encodes the Vif protein). We found the wtHIV grew to approximately similar levels regardless of whether or not any A3 was expressed in these cells (Figs 5D and S2). We also collected cell lysates from day 14 in the HIV-1ΔVif and wtHIV-1 infection and performed a western blot to examine the intracellular expression levels of A3G and A3C-A3H. We found that when we compared the intracellular expression in each of 3 biological replicates of the cell lines that were infected with wtHIV-1 to HIV-1ΔVif, the A3 expression was lower in the presence of HIV-1 Vif, consistent with the Vif-mediated degradation of A3 (Fig 5E). Together, these data show that A3C-A3H is just as potent of a restriction factor as A3G in the absence of HIV-1 Vif; however, it is nonetheless antagonized by Vif and targeted for degradation.

## Discussion

Here we combined two single domain A3 proteins, A3C and A3H that encode A3Z2 and A3Z3 domains, respectively, into a single molecule to test the hypothesis that there is novel antiviral potential in the human *A3* locus. We found that these A3C/A3H double domains can create super restriction factors with antiviral potency that is at least as potent as A3G both in single-cycle assays and in spreading infections of T cells. The ability of the A3C/A3H synthetic double domain proteins to inhibit reverse transcription after viral infection of the target cells, rather than an increased ability to induce hypermutation, correlates with their increased ability to inhibit HIV-1 (Fig 3). Thus, it is possible to create novel combinations of A3 domains with just as potent antiviral activity as A3G that enhance a non-enzymatic mechanism of action.

### Why are A3C/A3H double domains so potently antiviral?

Not all novel double domain A3 combinations gain such potent antiviral activity. We previously showed that linking together two A3H haplotypes to form an A3H-A3H double domain does not increase antiviral activity [42,43] and linking two A3C domains together to form A3C-A3C leads to modest increases in antiviral activity (Fig 1 and [22]). However, here we created heterologous double domains using A3C and A3H and show that A3C/A3H double domains are over 100-fold more potent than A3C-A3C (Fig 1B). One possibility is that combining two different evolutionary distinct domains, such as with A3G being a combination of Z2 and Z1 domains, creates a more potent restriction factor because each domain has specialized contributions to substrate specificity, binding affinity, deamination activity, and/or packaging into virions, allowing for independent and additive activities [10]. For example, both A3F and A3G primarily rely on only the C-terminal domains for catalytic activity, allowing for the N-terminal domain to perform other aspects involved in restriction [44–46]. The full-length human A3G structure provides insights about how the two domains interact to form a channel between the N-terminal and C-terminal domain to form additional affinity to ssDNA [47,48]. We speculate that having two different Z domains in a double deaminase domain protein could provide fitness advantages to sub-specialization of each domain.

A3C/A3H double domains, unlike their single domain counterparts or even A3G, restrict HIV-1 primarily through a deaminase-independent mechanism (Fig 3A and 3B). Our data is consistent with the model that these double domains have gained an ability to interfere with the reverse transcription process, leading to fewer intact HIV-1 integration products (Fig 3C).

The deaminase-independent mechanism of inhibition of HIV-1 could result from cumulative delays in reverse transcriptase products because of binding the template RNA, binding to negative-strand DNA, and/or binding to reverse transcriptase, thereby preventing proviral DNA synthesis [29–31,49–51]. A3F and A3G have been shown to interact with reverse transcriptase to negatively regulate its activity [28–31,51]. Additionally, dimerization of A3G has been shown to slow the dissociation of A3G from ssDNA and reduce its scanning ability [52]. A3H requires a double-stranded RNA to make functional dimers [33–35], but here we show that the binding affinity of the A3C-A3H double domain to the HIV-1 5'UTR RNA is over 10-fold greater than A3H$_{hap\ II}$ and over 200-fold greater than A3G (Fig 4). Thus, these data are consistent with the hypothesis that increased packaging as well as increased affinity for RNA compared to their single domain counterparts is the mechanism by which these super restriction factors block reverse transcriptase from synthesizing full-length proviral DNA.

The major factor mediating interactions between A3s and nucleic acids is the electrostatic interactions between positively charged amino acids and the negatively charged nucleic acid phosphate backbone. For A3H$_{hap\ II}$, the overall charge of the amino acid sequence at pH 7.5 is +6.3 (http://protcalc.sourceforge.net/). In contrast, A3C has an overall negative charge of -0.7. We hypothesize that the positive charge of the A3H$_{hap\ II}$ would increase the interaction time of A3C with RNA, similar to what has been shown for the two A3G domains on RNA and ssDNA [53,54]. However, the two domains of A3G are more different in charge with the N-terminal being +9.9 and the C-terminal domain being -6.3 at pH 7.5. This correlates with weaker RNA binding for A3G in comparison to A3C-A3H (Fig 4), which has less disparity between the charge of the two domains.

Furthermore, RNA binding has been implicated for proper subcellular localization of A3F, A3G, and A3H [33,34,55–61]. Treatment with RNase A can disrupt interactions between A3s and cellular proteins, hinting at RNA playing an important role in regulating A3 activities [33–35,62]. A3H$_{hap\ I}$ has been reported to have reduced RNA binding [62] and could explain why A3C/A3H$_{hap\ I}$ double domains are less active compared to A3C/A3H$_{hap\ II}$ double domains (Fig 1B). Additionally, as there is a gain in the protein expression level of A3H$_{hap\ I}$ in the A3C/A3H$_{hap\ I}$ double domains (Fig 1B), this phenotype could be due to an increase in RNA interactions leading to the gain in stability of A3H$_{hap\ I}$ chimeras.

## A3C-A3H inhibits HIV-1ΔVif, but not wtHIV-1 in a spreading infection in T cells

It has been argued that transient expression of A3 proteins in 293T cells exaggerates their antiviral activity and that stable expression in T cells is a better predictor of their true antiviral activity [14]. Here, we tested cells expressing A3C-A3H against wtHIV-1 and HIV-1ΔVif in spreading infections in Jurkat cells that stably expressed A3C-A3H and found that these experiments recapitulated the single-cycle infections, demonstrating that A3C-A3H is as potent as A3G in inhibiting HIV-1ΔVif, but though a novel, non-catalytic mechanism (Figs 1 and 3). We previously found that an A3C-A3C synthetic tandem domain protein was relatively resistant to Vif antagonism [22]. However, in contrast to the HIV-1ΔVif infection, we found that wtHIV-1 (i.e. HIV-1 that expressed Vif) was able to replicate in A3C-A3H-expressing cells to similar levels to the A3G- and no A3-expressing cells (Fig 5D). Thus, despite A3C-A3H being a novel target for HIV-1 Vif, this data shows that HIV-1 Vif has the potential to target novel A3 double domains. Vif uses three interfaces to bind to antiviral A3s: one for A3G, another for A3H, and a third for A3C/A3D/A3F [63]. In the double domain A3C/A3H combinations, either the A3C or the A3H determinants for Vif degradation must still be surface-exposed.

Future experiments will determine if we can additionally select for potent super restrictor A3 combinations that are resistant to Vif.

## Why has a A3Z3 rarely been used in a double domain A3?

Despite the potent antiviral activity of A3C/A3H double domains, no primate genome currently contains a functional double domain A3 containing a Z3 domain [8,9,12]. Interestingly, no mammalian groups have a detectable Z3 duplication except for in Carnivora, in which the *A3Z3* duplication has been almost entirely pseudogenized [9] (although felines make a splice RNA variant that encodes a readthrough product of a Z2-Z3 fusion protein [24]). These results suggest that Z3 domains may have alternative, harmful deaminase targets like cellular genomic DNA, precluding their inclusion in highly potent double domain A3s. A3B and A3H$_{hap\ I}$, A3s with more nuclear localization, have been implicated in contributing to cancer, suggesting that they could be detrimental to the host [64,65]. However, we were able to create cell lines that express A3C-A3H (Fig 5) with no obvious growth defects, although this does not rule out long-term or more subtle growth defects.

Another hypothesis is that generation of a cytidine deaminase-independent mechanisms of inhibition is inherently less optimal than an antiviral activity based on hypermutation, and that nature has selected against those A3 combinations of domains that do not favor enzymatic activity rather than inhibition of reverse transcription. Since A3-mediated hypermutation leads to broad and permanent inactivation of the viral genome, this mechanism of inhibition might have been selected. However, because cells expressing A3C-A3H where able to inhibit viral replication to similar levels as A3G, this possibility seems less likely. Nevertheless, by making novel tandem domain proteins not found in primate genomes, we have learned that more potent antiviral activity can be achieved with deaminase-independent mechanisms, such as increase packaging of A3s into budding virions, increased RNA binding, and inhibition of reverse transcription. Thus, our data suggest that there is an untapped mechanism of potent antiviral activity within the *A3* locus that could block reverse transcription directly rather than act through hypermutation.

## Materials and methods

### Plasmid constructs

The plasmids were created using the *A3C* sequence [15] and *A3H$_{hap\ I}$* [18] and were designed based on similar alignments as A3D and A3F, incorporating the naturally found short linker amino acid sequence between both domains Arg-Asn-Pro (RNP) in A3D and A3F as previously described [22]. Hybrid *A3* constructs generated via gene synthesis (Integrated DNA Technologies, IDT) for both *A3C-A3H$_{hap\ I}$* and *A3H$_{hap\ I}$-A3C*. To create all mutations and the *A3H$_{hap\ II}$* variants of these *A3C/A3H* double domains, Site-Directed Mutagenesis using the QuikChange II XL kit (Agilent, #200522–5) was performed and the mutations were confirmed by sanger sequencing. To convert the *A3H$_{hap\ I}$* into *A3H$_{hap\ II}$*, the following mutations were made: G105R and K121D, using Site Directed Mutagenesis [18]. Wild-type A3H$_{hap\ II}$ behaves the same as A3H$_{hap\ I\ R105\ D121}$ [42]. To create the active site knockout mutants, mutations were made in both the N- and C- terminal domains, E68A and E254A, for a catalytically inactive variant. All constructs have a C-terminal 3XFLAG epitope tag and were cloned into the pcDNA4/TO vector backbone (Thermo Fisher, #V102020) using restriction sites at EcoRI/XhoI.

### Cell culture and transfections

Jurkat (ATCC TIB-152) and SUPT1 (ATCC CRL-1942) cells were maintained in RPMI Medium (Gibco, #11875093), with 10% Fetal Bovine Serum (GE Healthcare, #SH30910.03), 1%

Penicillin Streptomycin (Gibco, #15140122), and 1% HEPES at 37˚C, referred to as RPMI complete. HEK293T cells (ATCC CRL-3216) were maintained in Dulbecco's modified Eagle's medium (Gibco, #11965092) with 10% Fetal Bovine Serum (GE Healthcare, #SH30910.03), and 1% Penicillin Streptomycin (Gibco, #15140122) at 37˚C. The plasmids were transfected into the cells using TransIT-LT1 transfection reagent (Mirus, MIR2304) at a ratio of 3:1 mirus:plasmid.

### Single-cycle infectivity assays

Single-cycle infectivity assays were previously described in [20,22]. In short, 293T cells were seeded in 6-well plates at a density of $1.5 \times 10^5$ cells per mL. 24 hours later, the cells were transfected with 600ng of *HIV-1ΔEnvΔVif* provirus (LAI strain), 100ng of *L-VSV-G*, and 100ng of *A3* plasmid for all single-cycle infectivity assay unless otherwise noted. 72 hours post transfection, virus was collected and filtered through a 0.3 micron syringe filter. Virus titers were determined using a SG-PERT assay as described in [40]. $2 \times 10^4$ SUPT1 cells per well were seeded into a 96 well plate supplemented with 20μg/mL of DEAE-dextran. All infections were done in technical triplicate. 72 hours post infection, the cells were lysed with a 1:1 ratio of virus to Bright-Glo luciferase assay media (Promega, #E2610) and the contents were analyzed on a luminometer (LUMISTAR Omega, BMG Labtech). Values were normalized to the no A3 samples and graphed on Prism software.

### Western blotting

Cells were lysed with NP-40 buffer (0.2M Sodium Chloride, 0.05M Tris pH7.4, 0.5M NP-40 Alternative, 0.001M DTT, Protease Inhibitor Cocktail (Roche Complete Mini, EDTA-free tablets, 11836170001) 72 hours post transfection. The cell lysates were centrifuged at 4˚C at 1,000 rpm for 10 minutes to remove the nuclei pellet. The supernatant was transferred to a new set of a tubes and spun down at 13,000 rpm for 10 minutes at 4˚C to remove the remaining debris. The supernatant was transferred to a new set of tubes and lysed in 4X loading dye (Invitrogen, #NP0007) and boiled at 95˚C for 10 minutes. The boiled samples were resolved on a 4–12% Bis-Tris gel, transferred to a nitrocellulose membrane (Bio-Rad, 1620115), and blotted with antibodies to detect protein levels. Anti-FLAG (Sigma, F3164), and anti-p24gag (NIH-ARP, 3537) antibodies were used at 1:10,000, and Actin (Sigma, A2066), StarBright Blue 520 Goat Anti-Rabbit IgG (Bio-Rad, 12005869) and StarBright Blue 700 Goat Anti-Mouse IgG (Bio-Rad, 12005866) were used at a ratio of 1:5,000.

### Quantification of late reverse transcription products

Quantification of late reverse transcription products was previously described in [22,66]. In short, cells were harvested 19 hours post-infection and unintegrated cDNA was collected using the Qiagen mini-prep kit (QIAprep Spin Miniprep Kit, # 27106). Samples were concentrated using the Zymo DNA Clean and Concentrator-25 kit (Zymo, D4033). HIV cDNA was amplified with TaqMan gene expression master mix (Applied Biosystems, 4369016), J1 FWD (late RT F)—ACAAGCTAGTACCAGTTGAGCCAGATAAG, J2 REV (late RT R) GCCGTG CGCGCTTCAGCAAGC, and LRT-P (late RT probe)—6-carboxyfluorescein (FAM)-CAGTG GCGCCCGAACAG GGA-6-carboxytetramethylrhodamine (TAMRA) [67,68]. Data were acquired on an ABI QuantStudio5 real-time (qPCR) machine and analyzed on Prism software.

### A3 mediated hypermutation assay

The A3 mediated hypermutation assay was previously described in [22]. In short, SUPT1 cells were infected with Benzonase-treated HIV-1ΔVifΔEnv virus pseudotyped with VSVg and the

designated A3. 19 hours later, unintegrated viral cDNA was isolated using a Qiagen miniprep kit (QIAprep Spin miniprep kit; catalog no. 27106). To determine A3-mediated mutations, we used a barcoded Illumina deep-sequencing approach as previously described [22,69]. Samples were amplified, quantified, pooled, purified via gel electrophoresis, and sequenced on an Illumina MiSeq sequencer, using 2x250 paired-end reads.

The dms_tools2 software packaged was used to align sequencing reads and build consensus sequences for each uniquely tagged DNA molecule [70]. Error-corrected reads were compared to the target sequence to determine the number, identity, and surrounding nucleotides of all substitutions in each read. Reads with high numbers of substitutions (>10% of non-G nucleotides) at the junction of the two paired-end reads were removed from the analysis as these substitutions were most often found to be alignment artifacts. Since A3s are known to cause G-to-A substitutions, we subsampled our data to specifically at G-to-A substitutions. The data is shown as the frequency of reads in each sample with a given number of G-to-A mutations (0, 1, 2, etc., up to 9 and then 10+).

## Jurkat T cell lines stably expressing A3 proteins

In order to create constitutively expressed A3 cell lines, the Sleeping Beauty transposase system [39] was adapted to electroporate Jurkat cells using the Lonza SE Cell Line kit (Lonza, V4SC-1960). The pSBbi-RP plasmid was a gift from Eric Kowarz (Addgene plasmid #60513) [71]. *A3G-3XFLAG* was cloned into the *pSBbi-RP* vector using the NcoI/XbaI restriction sites. *A3C-A3H_hap II-3XFLAG* was cloned into the *pSBbi-RP* vector using the EcoRV/XbaI restriction sites. Using the Lonza 4D-Nucleofactor (program 'Jurkat E6.1(NEW)' and pulse code 'CK116'), Jurkat cells were electroporated with the *pSBbi-RP-A3* and *pCMV(CAT)T7-SB100* (gift from Zsuzsanna Izsvak, Addgene plasmid #34879) [72]. Post electroporation, cells were recovered in RPMI and 24 hours later, transferred into RPMI supplemented with 0.4μg/mL puromycin for selection. To further ensure that only electroporated cells survived, cells were flow sorted for dTomato positive cells and maintained in RPMI media supplemented with 0.2μg/mL puromycin (Sigma, #P8833) selection. Additionally, cells were sorted for similar CD4 levels, using APC anti-human CD4 (PharMingen, #555349), based on the CD4 levels of the A3C-A3H_hap II expressing cells.

## Spreading infection

wtHIV-1 (LAI strain) and HIV-1ΔVif virus stocks were created by transfecting HEK293T cells with 1μg of viral plasmid per well in a 6 well plate. The virus was titered on the Jurkat stable cell lines, via flow cytometry staining for p24-FITC (Beckman Coulter, 6604665) positive cells. The stable Jurkat cell lines were then infected at an MOI of 0.01 and 0.05. Virus and 20μg/mL DEAE-Dextran in RPMI were added to cells, spinoculated for 30 min at 1100xg, and post infection, fresh RPMI media was added to the cells. The spreading infection was drawn out for 14 days and performed in triplicate. The cells were closely monitored, and supernatant samples were taken every 2–3 days. Reverse transcriptase was quantified in the collected viral supernatant using a SG-PERT assay [40]. Cell lysates were collected on day 14 and split into two samples. These cell lysates were run on a western blot to check for A3 protein expression and used to harvest gDNA (QIAamp DNA Blood mini kit, #51104) for the A3 mediated hypermutation assay.

## Steady-state rotational anisotropy

For generation of RNA *in vitro*, the HIV 5′-UTR (nucleotides 1–497) was cloned into pSP72 vector (Promega) using BglII and EcoRI sites under the control of T7 promoter. All constructed

plasmids were verified by DNA sequencing. Primers were obtained from IDT and are reported in Feng et al [73]. Fluorescently labeled RNA was produced by transcribing pSP72 DNA cut with EcoRI *in vitro* using T7 RNA polymerase with a nucleotide mixture containing fluorescein-12-UTP (Roche Applied Science). Steady-state rotational anisotropy reactions (60 μL) were conducted in buffer containing 50 mM Tris, pH 7.5, 40 mM KCl, 10 mM $MgCl_2$, and 1 mM DTT and contained 10 nM fluorescein-labeled 5'UTR RNA and increasing amounts of A3 ($A3H_{hap II}$, 0.1–61 nM; $A3C\text{-}A3H_{hap II}$, 0.0045–6.040 nM; and A3G, 0.36202 nM). A Quanta-Master QM-4 spectrofluorometer (Photon Technology International) with a dual emission channel was used to collect data and calculate anisotropy. Measurements were performed at 21˚C. Samples were excited with vertically polarized light at 495 nm (6-nm band pass), and vertical and horizontal emissions were measured at 520 nm (6-nm band pass). The $K_d$ was obtained by fitting to a hyperbolic decay curve equation using SigmaPlot version 11.2 software.

## Supporting information

**S1 Fig. Annotated $A3C\text{-}A3H_{hapI}$ and $A3H_{hapI}\text{-}A3C$ sequences.** Full length sequences used to construct the A3C-A3H and A3H-A3C double deaminase domains. The "RNP" amino acid sequence (Arginine-Asparagine-Proline) that links the two deaminase domains together is highlighted in yellow. The essential glutamic acid that is necessary for deaminase activity is in red text. The A3H human polymorphic sites are shown in blue text.
(TIF)

**S2 Fig. A3C/A3H expressing cells inhibit viral replication at an MOI = 0.05.** Spreading infection kinetics of a replication-competent HIV-1 with a deletion that spans the Vif open reading frame (called HIV-1ΔVif) (A) or wtHIV-1 (LAI isolate) (B). The Jurkat cells expressing no A3 (circle, grey line), A3G (square, orange line), or $A3C\text{-}A3H_{hap II}$ (celled A3C-A3H, triangle, blue line) were infected at a low MOI (MOI = 0.05) in triplicates. Virus production was monitored over time by collecting supernatant and measuring RT activity (mU/mL) using a SG-PERT assay. Error bars represent the standard error across the 3 biological replicates.
(TIF)

## Acknowledgments

We thank Molly Ohainle, Jeannette Tenthorey, and Nicholas Chesarino for their helpful feedback on this manuscript, James Dargan for his assistance with the A3-mediated hypermutation code, and the Fred Hutchinson Shared Resources Genomic Core.

## Author Contributions

**Conceptualization:** Mollie M. McDonnell, Linda Chelico, Michael Emerman.

**Data curation:** Mollie M. McDonnell, Linda Chelico.

**Formal analysis:** Mollie M. McDonnell, Linda Chelico.

**Funding acquisition:** Michael Emerman.

**Investigation:** Mollie M. McDonnell, Suzanne C. Karvonen, Amit Gaba, Ben Flath.

**Methodology:** Mollie M. McDonnell, Suzanne C. Karvonen, Amit Gaba, Ben Flath.

**Project administration:** Linda Chelico, Michael Emerman.

**Supervision:** Linda Chelico, Michael Emerman.

**Validation:** Suzanne C. Karvonen.

**Visualization:** Mollie M. McDonnell.

**Writing – original draft:** Mollie M. McDonnell, Suzanne C. Karvonen, Linda Chelico, Michael Emerman.

**Writing – review & editing:** Mollie M. McDonnell, Suzanne C. Karvonen, Linda Chelico, Michael Emerman.

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
