## [Decision Letter · Decision Letter 0]

10 May 2021

Dear Dr. Emerman,

Thank you very much for submitting your manuscript "Highly-potent, synthetic APOBEC3s restrict HIV-1 through deamination-independent mechanisms" for consideration at PLOS Pathogens. As with all papers reviewed by the journal, your manuscript was reviewed by members of the editorial board and by several independent reviewers. The reviewers appreciated the attention to an important topic. Based on the reviews, we are likely to accept this manuscript for publication, providing that you modify the manuscript according to the review recommendations.

Sincerely,

Welkin E. Johnson

Associate Editor

PLOS Pathogens

Richard Koup

Section Editor

PLOS Pathogens

Kasturi Haldar

Editor-in-Chief

PLOS Pathogens

orcid.org/0000-0001-5065-158X

Michael Malim

Editor-in-Chief

PLOS Pathogens

orcid.org/0000-0002-7699-2064

Reviewer Comments (if any, and for reference):

Reviewer's Responses to Questions

**Part I - Summary**

Reviewer #1: In this manuscript, the authors explore combining deaminase domains from individual APOBEC proteins to determine whether this would increase antiviral activity against HIV. The double deaminase domain protein APOBEC3G is the most potent of the APOBECs in restricting HIV, and it is currently unknown whether all of the possible deaminase domain combinations have been evolutionarily explored in nature. The authors determine that combining deaminase domains from APOBEC3C and 3H results in a super-restrictor that limits HIV replication at least as well as or better than APOBEB3G depending on which assay they utilize. They mechanistically determine that the strong antiviral activity is a result of both deaminase activities and enhanced interaction of the novel restriction factor with viral RNA, correlating with increased packaging into virions. Though the practical/translational impacts of the work are unclear, the experiments are very well done, the data are presented clearly, and the results are informative with regard to the evolutionary history and the breadth of evolutionary sampling of possible antiviral APOBECs.

Reviewer #2: In this work the authors demonstrated that a synthetic A3C/A3H di-domain chimera acquired more efficiency in virion packaging and enhanced antiviral activity, which is comparable to the most potent natural di-domain A3G, and the antiviral activity is conferred by both deaminase-dependent and independent mechanisms. The authors also showed that the deaminase-independent activity is correlated to the increased RNA binding affinity of the A3C/A3H chimera. This is a clean work with only minor modifications needed.

1. Figure 4 only shows the anisotropy measurements for A3C-A3H, A3H and A3G. How about the others, e.g., A3H-A3C and A3C?

2. In Figure 5C the No-A3 control has considerable level of G-to-A mutations. Why is it so different from that in Figure 3A?

**Part II – Major Issues: Key Experiments Required for Acceptance**

Reviewer #1: Figure 2 is representative of 3 experiments. Can the results from the 3 experiments be quantified and included as a bar graph?

The usefulness of the research is not entirely clear. The authors have coined the term “super-restrictor,” and their novel proteins could potentially be useful in future gene therapy approaches to HIV treatment. However, all of the assays are done with HIV-deltaVIF, which is sensitive to APOBEC inhibition. Creating an APOBEC-based super-restrictor that evades VIF inhibition and can thus inhibit WT HIV-1, would be a more exciting advance. Based on published literature regarding the VIF-APOBEC interaction interferface, can the authors create and test such a version of the A3H/A3C constructs?

Reviewer #2: (No Response)

**Part III – Minor Issues: Editorial and Data Presentation Modifications**

Reviewer #1: (No Response)

Reviewer #2: (No Response)

PLOS authors have the option to publish the peer review history of their article (what does this mean?). If published, this will include your full peer review and any attached files.

Reviewer #1: No

Reviewer #2: No

Figure Files:

Data Requirements:

Reproducibility:

References:

---

## [Editor Report · Decision Letter 1]

7 Jun 2021

Dear Dr. Emerman,

We are pleased to inform you that your manuscript 'Highly-potent, synthetic APOBEC3s restrict HIV-1 through deamination-independent mechanisms' has been provisionally accepted for publication in PLOS Pathogens.

Best regards,

Welkin E. Johnson

Associate Editor

PLOS Pathogens

Richard Koup

Section Editor

PLOS Pathogens

Kasturi Haldar

Editor-in-Chief

PLOS Pathogens

orcid.org/0000-0001-5065-158X

Michael Malim

Editor-in-Chief

PLOS Pathogens

orcid.org/0000-0002-7699-2064
---

## [Editor Report · Acceptance letter]

22 Jun 2021

Dear Dr. Emerman,

We are delighted to inform you that your manuscript, "Highly-potent, synthetic APOBEC3s restrict HIV-1 through deamination-independent mechanisms," has been formally accepted for publication in PLOS Pathogens.

Best regards,

Kasturi Haldar

Editor-in-Chief

PLOS Pathogens

orcid.org/0000-0001-5065-158X

Michael Malim

Editor-in-Chief

PLOS Pathogens

orcid.org/0000-0002-7699-2064